# Brain development dictates energy constraints on neural architecture search: cross-disciplinary insights on optimization strategies

## Abstract

Today's artificial neural architecture search (NAS) strategies are essentially prediction-error-optimized. That holds true for AI functions in general. From the developmental neuroscience perspective, I present evidence for the central role of metabolically, rather than prediction-error-, optimized neural architecture search (NAS). Supporting evidence is drawn from the latest insights into the glial-neural organization of the human brain and the dynamic coordination theory which provides a mathematical foundation for the functional expression of this optimization strategy. This is relevant to devising novel NAS strategies in AI, especially in AGI. Additional implications arise for causal reasoning from deep neural nets. Together, the insights from developmental neuroscience offer a new perspective on NAS and the foundational assumptions in AI modeling.

## 1   Introduction

This work is written by a neuroscientist, not a computer scientist. Apologies in advance for any naïveté. This article is theoretical in nature and based on evidence from developmental neuroscience. The aim is to generate new testable hypotheses about the design of artificial intelligence (AI) from the physiological point of view. The manuscript is organized as follows. First, I briefly review the general process of the neural architecture search (NAS). Second, I discuss the neuroanatomical and neurophysiological evidence for glial-neural network architecture. Third, I add evidence for the fundamental role of metabolic constraints in such brain architectures for brain development, i.e., biological NAS, and brain function, i.e., multimodal multitask deep neural networks. Fourth, from the perspective of physics, we consider the dynamic coordination theory which provides a mathematical bridge between the energy-driven coordination of weakly coupled non-linear oscillators and brain networks. Finally, we synthesize these insights into a revised version of the NAS process. The manuscript is concluded with a discussion of limitations and future directions.

## 2   From artificial to brain-inspired NAS

### 2.1   Neural architecture search: status quo

For the context of the present work, it is important to summarize the steps of NAS. I reproduce these steps as defined by Elsken et al. [2019]. The credit for these steps goes entirely to the cited authors. Briefly, the search space is defined and then explored using search strategies in an iterative manner as part of the performance estimation strategy.

Submitted to 37th Conference on Neural Information Processing Systems (NeurIPS 2023). Do not distribute.

1. The search space defines which architectures can be represented in principle. Incorporating prior knowledge about typical properties of architectures well-suited for a task can reduce the size of the search space and simplify the search. However, this also introduces a human bias, which may prevent finding novel architectural building blocks that go beyond the current human knowledge.

2. The search strategy details how to explore the search space. In the conventional view of NAS, the search space can be exponentially large or even unbounded, while in the proposed energy-constrained view of NAS, this is not the case. It encompasses the classical exploration-exploitation trade-off since, on the one hand, it is desirable to find well-performing architectures quickly, while on the other hand, premature convergence to a region of suboptimal architectures should be avoided.

3. The conventional objective of NAS is typically to find architectures that achieve high predictive performance on unseen data. Performance estimation represents the process of estimating the predictive performance of a deep neural net. The simplest option is to perform standard training and validation of the architecture on data, but this is unfortunately computationally expensive and limits the number of architectures that can be explored. Much recent research, therefore, focuses on developing methods that reduce the cost of these performance estimations.

In each of these three steps, I show in the following how the energy-driven glial-neuronal NAS approach can provide useful constraints. To do that, we first need to review the evidence from neuroscience and dynamic coordination theory that formalizes mathematically the relationship between energy optimization and network communication.

## 2.2 Energy-constrained glial-neural nets

The early notion that neurons compute to effectively reduce the prediction error does not hold true in biological brains, especially in the human brain on two accounts: first, neurons are dynamically and spatiotemporally organized with as numerous glial cells to perform their functions, and, second, these functions are energy-constrained first and, most likely, prediction-error-minimizing second. We observe the paramount role of metabolic optimization, i.e., neural nets develop and are sustained under energy constraints. It is then questionable that error prediction should be the primary objective of a deep neural net evolution and function. This has implications throughout the NAS. In the following, I review some evidence that biological parallels of deep neural nets are in fact glial-neural nets in which energy optimization drives the NAS.

First, neurons should not continue to be viewed as the sole foundation for biologically inspired AI. In fact, when it comes to brain development and function in health and disease, they are at best an equal partner of the glial cells, with a complex, spatiotemporally dynamic relationship between these two major cell categories. The estimates for the human brain glia-to-neuron ratio range between $1:1 - 6:1$ (von Bartheld et al. [2016], Herculano-Houzel [2014]). Of the 170.68 billion cells, 84.6 billion are glial cells and 86.1 billion are neurons (Collman [2022]). Moreover, from the perspective of artificial general intelligence (AGI), it is noteworthy that the ratio of glia to neurons is brain region specific. This ratio is evolutionarily conserved by brain regions across species that diverged as far as 90 million years ago.

This evidence indicates that task-specific glia-neuron ratios in deep neural networks are meaningful and fundamental while imposing constraints on brain function. With regard to synaptic complexity between glia cells and neurons, human astrocytes, the most populous glia cell, differ dramatically from astrocytes in other primates as well as in rodents: 2 million synapses versus 120,000 are found in the human astrocytes and the degree of this synaptic plasticity is driven by astrocytes-neuronal interactions in the developing brain (Cheng et al. [2023]).

Second, energy constraints, mediated via glia, enforce sparse neural coding, a property conserved evolutionarily (von Bartheld et al. [2016], Herculano-Houzel [2014]). Perhaps the most widely known example of this on a network level is the synaptic consolidation during sleep with some synapses being eliminated and some reinforced. On a cellular level, larger neurons show reduced rates of excitatory synaptic transmission.

Third, in developmental neuroscience, there is evidence that energy constraints also drive brain development and pathophysiologies underlying Autism Spectrum Disorder and neurodegenerative

conditions like Alzheimer's (Desplats et al. [2019], Frasch et al. [2019]). This is the subject of the next subsection.

## 2.3 Metabolic constraints on brain development: biological glial-neuronal NAS

Glial cells, such as astrocytes and microglia, participate in energy management, synaptic pruning, and plasticity of developing, adult, and aging brain. Glial functional alterations are hallmarks of the brains of people with Autism Spectrum Disorder or neurodegenerative conditions such as Alzheimer's (Desplats et al. [2019], Frasch et al. [2019]). Notably, in both conditions, the glial function itself is energy-dependent. The glial cells adapt to energy availability in early life which in turn reflects intrauterine adversity the fetal brain may experience (Desplats et al. [2019], Frasch et al. [2019]). Put differently, biological NAS can be seen as a process of spatiotemporal integration of early-life adversities and developmental programming, such as prenatal stress and the accompanying or independently occurring systemic- and neuro-inflammation, on glial energy reserves that modulate the risk for neurodegeneration via modulation of the pace or extent of immunosenescence, i.e., neural function and plasticity (Desplats et al. [2019]).

Such energy-driven network architecture selection suggests a template for NAS in AI. Conversely, artificial NAS that incorporates these relationships has the potential to yield insights into developmental neuroscience and the neuroscience of aging, especially the aforementioned neurological conditions.

This raises the question of how exactly one would go about redefining the relationship between the artificial neurons to include the metabolic cost of computation. To tackle this question, let us consider first another bit of evidence from integrative neuroscience.

## 2.4 Dynamic coordination during behavioral states points to metabolic optimization

Brain's behavioral states, such as the NREM and REM states, also known as quiet and active states in neonates, are known examples of state-specific system-wide energy management (Figure 1) (Schmidt [2014]). The relevant insight in Figure 1 is that the systemic metabolic states are also reflected in or driven by dynamic relationships between the participating oscillatory networks, the one generating heart beats fluctuations and the one responsible for fluctuations in breathing movements. These phenomena can be described mathematically in simple constructs of Farey trees or Arnold tongues: low energy state of quiet sleep is accompanied by a 3:1 ratio of dynamic coordination between heartbeats and breathing movements, while the higher energy state of active sleep is accompanied by a break-down of such ratios into those more off center (or higher hierarchy) of the ratio distribution (Hoyer et al. [2001], Gebber et al. [1997]). In the following section, we review the more general case of the theory of dynamic coordination which provides a generic mathematical formulation of the observed connection between energy consumption and network dynamics.

## 2.5 Dynamic coordination and metabolic optimization: linking meta-/multi-stability to the energy landscape

One could conceptualize energy-driven systems' optimization as the optimal solution to the "too many degrees of freedom" state of biological systems. In their 1988 Science paper, Schoener and Kelso express the solution via the following simple equation for a basic case of two weakly coupled nonlinear oscillators (Schöner and Kelso [1988]). A candidate collective variable that succinctly captures the dynamics of such coordinative patterns is the relative phase between the two rhythmically moving components. In the case of an artificial neural network, this could be a coordinative dynamics of two neurons or a glia-neuron pair. The collective variable $\phi$ (relative phase) describes the system's dynamic behavior on the energy landscape V as follows:

$$\phi = -\frac{dV(\phi)}{d\phi} + noise$$

The attractors are thus the minima of V, whereas the maxima of V are unstable fixed points that separate different basins of attraction. An intrinsic feature of this dynamic pattern approach is the invariance of function under a change of material substrate - a reconfiguration of the connections or couplings among "neural" elements (Schöner and Kelso [1988], Kelso [1995], Tognoli and Kelso [2014], Kelso [2012]). That is, the function is not rigidly coded into the neural network. This can encode multimodal and multitask capabilities while also optimizing for energy. Further reading can be found here: Tognoli and Kelso [2014], Kelso [2012], Chauhan et al. [2022], Kelso [2021]. In the

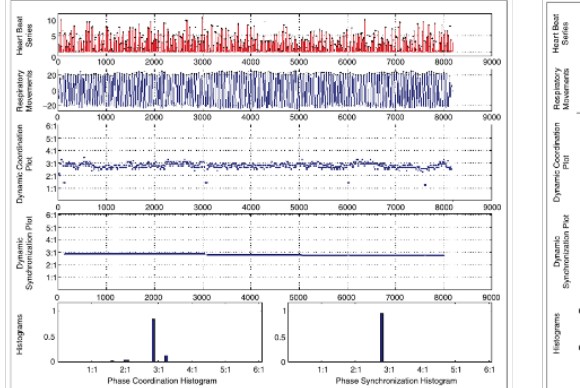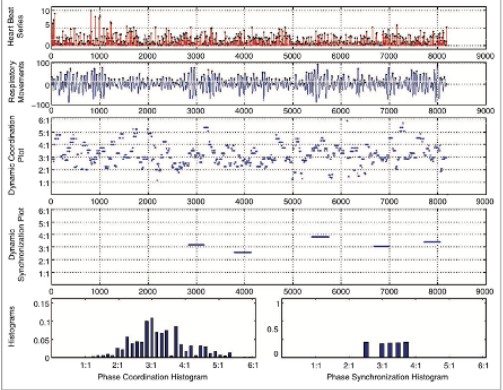

Figure 1: Relationship of the brain systems' energy states (low in quiet/NREM sleep; high in active/REM sleep), behavioral states (memory consolidation/synaptic pruning in quiet sleep; memory formation in active sleep)(Tononi and Cirelli [2020]) and mathematical properties of dynamic coordination expressed as Farey trees. Reproduced with permission from IEEE (Hoyer et al. [2001]). This is one example of many whereby biological complex open systems follow fundamental rules of dynamic coordination (Schöner and Kelso [1988]).

proposed context, the application of the dynamic pattern approach to modeling deep glial-neural networks can be seen as an extension of the well-established free energy principle, now constrained on the available energy for predictive coding (Kirchhoff et al. [2018], Friston and Kiebel [2009], Isomura et al. [2022]).

## 2.6 Revisiting neural architecture search: brain-inspired at every step

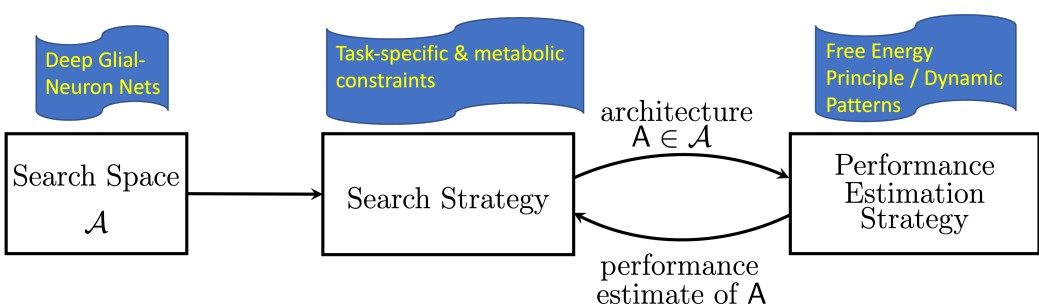

Figure 2: Augmentation of NAS strategy by introducing glial-neural network design with energy-constrained architecture search. Based on Elsken et al. [2019]

Based on the presented evidence from neuroscience, we can now update the NAS strategy accordingly (Figure 2). The search space can be re-defined as including glial-neuronal ensembles tuned to process information under metabolic constraints, rather than neuronal ensembles only tuned to reduce the prediction error. The search strategy now includes task-specific metastable regimes defined by a collective variable such as the relative phase $\phi$ with intrinsically encoded metabolic constraints. Lastly, the performance estimation strategy is driven by a combination of the free energy principle and dynamic coordination theory iteratively optimizing the performance estimates (minimum energy $E_{min}$, maximum information $I_{max}$) of the chosen architectures.

The novel perspective the developmental neuroscience contributes to this NAS strategy is the evolutionary, task- and modality-specific function of glial/neuronal distribution that is solved for during the Search Strategy step.

Such NAS strategy will yield deep glial-neural networks (DGNNs) which are multi-stable, hence capable of representing multiple behaviors with multistability arising from $E_{min}$ constraints to represent multitask/multimodality with $I_{max}$ predictive performance.

## 3  Limitations and future directions

Current AI is based on optimization strategies for information ($I_{max}$) without metabolic constraints ($E_{min}$). Developmental neuroscience is a natural yet seemingly neglected starting point for learning how to evolve energy- and computationally efficient and resilient pattern recognition in AI, including AGI.

In neuroscience, the free energy principle and the dynamic coordination theories provide mathematical formalisms to bridge this gap. Explicit metabolic constraints minimize surprisal maximizing predictive performance and optimizing energy utilization for information processing.

Neurons are but part of the family of brain cells involved in this process in

a) space: different brain regions specialize in different tasks, albeit this is dynamically regulated; and

b) time: as a function of developmental cues and allostatic load - metastability.

Future research should leverage deeper glial-neural networks. Some precedents exist (Mesejo et al. [2015]).

We need NAS algorithms incorporating $E_{min}$ x $I_{max}$. Is $E_{min}$ for AI just the cost of electricity or also an intrinsic system's property as exemplified by the theory of dynamic coordination? Evolutionary considerations and observations from developmental neuroscience suggest the latter should be considered if we are to further learn from biological brains. Future research will address the question if the collective variable $\phi$ (relative phase) can augment (multi-step or quantum?) or replace the traditional weights.

Related to the above question of phase versus weight optimization is the question if $E_{min}$ is more "important" than $I_{max}$. Modeling will be able to address this question.

What are the implications of biomimetic NAS design for generative DGNNs behavior? Can such architecture help design safer AGIs?

We started out with biological brains, let's return to them. The order parameters identified in multistable DGNNs can inform a new generation of neuroscience models and experiments. Moreover, causal explainable DNNs for modeling brain behavior would be possible thanks to such intrinsically energy-driven NAS design.

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
