# OpenReview forum: "Brain development dictates energy constraints on neural architecture search: cross-disciplinary insights on optimization strategies"
_NeurIPS.cc/2023/Conference — Submitted to NeurIPS 2023_

### Official Review · Reviewer_i8bJ · 2023-07-04

**Soundness:** 2 fair
**Presentation:** 2 fair
**Contribution:** 2 fair
**Rating:** 3
**Confidence:** 3

**Summary:**

This is a theoretical paper that proposes novel ideas for neural architecture search based on neuroscientific research. The author argues for the fundamental role of metabolic contains in the development of biological neural networks, which should inform the optimization strategies used for artificial neural networks.

**Strengths:**

- The paper points out interesting and neglected constraints that should be investigated for NAS

**Weaknesses:**

- The authors do not present any experiments that could strengthen their hypothesis
- The paper could give a more comprehensive overview of the current state of NAS, also including early work from the evolutionary community that already did incorporate energy constraints in the form of connection costs, e.g. "Clune, Jeff, Jean-Baptiste Mouret, and Hod Lipson. "The evolutionary origins of modularity." Proceedings of the Royal Society b: Biological sciences 280.1755 (2013): 20122863."
- Some of the concept, such as glia-neural networks, could be explained in more detail


**Questions:**

What would be concrete experimental setups that can be used to test the hypothesis proposed in this paper?

Currently, it is hard to judge how useful and impactful this paper would be for the wider machine-learning community. Initial proof-of-concept results would go a long way in pursuing others about this new proposed direction for NAS. Even though the presented ideas are interesting, In its current form, the paper seems better suited as a workshop paper and not quite ready for the main conference.

**Limitations:**

Yes.

---

> ### Author Rebuttal · Authors · 2023-08-09
>
> The primary impact for the ML community would be NAS strategies that yield more energy-efficient (neural-glial) architectures, potentially with novel emergent properties. This is relevant for both cloud and edge implementations.
>
> I thank the reviewer for the thoughtful and helpful comments and criticisms.
> The present manuscript shares ideas that are intended to be a transdisciplinary observation and a thought experiment.
> It is designed to motivate the NAS community, or the AI community in general, to consider the current constraints in AI design against what developmental neuroscience is pointing to as 1) a broader context of biologically inspired NAS and 2) to start conceiving concrete ways to incorporate the ideas presented herein for testing in silico.
> Some general, novel, optimization strategies are presented that can be implemented to that end.
>
> It was not the objective of this work to provide an overview of NAS, since this is, as stated at the onset, not the field of deep expertise by the authors.
>
> I appreciate the reference to https://doi.org/10.1098/rspb.2012.2863 - a brilliant paper that does incorporate the notion of energy constraints! The authors do not however make the connection to AI design nor discuss the role of glial cells as is attempted in the present work. I agree this work should be cited here and will revise it accordingly.
>
> I agree and appreciate the comment that the concept of glial-neural networks could be expanded upon. In the interest of space and focus, it was felt that the present level of depth with key literature references suffices for the stated objective of the manuscript. I would like to leave the transfer of this idea into NAS space to future work. The key message is that such a transfer is worthwhile. It is interesting to consider whether a whole new computation or optimization unit (in contrast to an artificial neuron) is required to model glial-neuronal interaction. It is possible and likely that several solutions exist which could be tested computationally in a systematic way.
>
> With these comments in mind, the primary impact for the ML community would be NAS strategies that yield more energy-efficient (neural-glial) architectures, potentially with novel emergent properties (as we have witnessed with LLMs). There are implications here both for cloud solutions as well as for AI training on edge.
>
> I recognize that such constraints may severely limit the manuscript in its current form from being considered for the NeurIPS. I thank you again the reviewer for the comments which will help with moving this work forward.

---

> > ### Comment · Reviewer_i8bJ · 2023-08-15
> >
> > Thank you for the clarifications. My main concerns still hold so I won't change my score right now but I'm very much looking forward to seeing experimental results of these interesting ideas in the future.

---

### Official Review · Reviewer_t3j3 · 2023-07-08

**Soundness:** 1 poor
**Presentation:** 1 poor
**Contribution:** 1 poor
**Rating:** 2
**Confidence:** 3

**Summary:**

This paper proposes an approach to neural architecture search that optimizes for energetic efficiency and draws inspiration from the role of glial cells in biological brains.

**Strengths:**

The idea of energetic efficiency objective functions for neural architecture search may have merit.

**Weaknesses:**

The paper does not really make a technical contribution -- there are no experiments or theoretical results.  The paper is essentially a proposal for a high-level idea about neuroscience-inspired neural architecture search, which is not adequate for a conference like NeurIPS.  Morevoer, the survey of neuroscience literature, while potentially interesting, is not presented in a way that would be comprehensible or actionable for machine learning researchers hoping to leverage these ideas.  I encourage the author(s) to make more concrete their proposed approach and evaluate it empirically if their goal is to improve upon current NAS methods.

**Questions:**

Could the authors describe in detail (e.g. with equations or an algorithm) a proposed NAS method using their ideas?

**Limitations:**

I found the discussion of limitations hard to follow and so cannot evaluate it properly.

---

> ### Author Rebuttal · Authors · 2023-08-09
>
> We thank the reviewer for the comments.
> We hope the responses to the other two reviewers' comments also address the concerns of this reviewer.
> We outlined the skeleton of this in Figure 2 and on Line 128.
> It has been the intention of this contribution to attract interested members of the AI/NAS community present at NeurIPS to do just that.

---

> > ### Comment · Reviewer_t3j3 · 2023-08-18
> >
> > Thank you for the responses.  While I agree this topic may be of interest to the NeurIPS community and encourage the authors to continue their work / discuss it with others in the community, I maintain my assessment that this manuscript is not adequate for publication in the conference.  I look forward to seeing experimental follow-ups to the authors' proposals in the future.

---

### Official Review · Reviewer_kaEY · 2023-07-13

**Soundness:** 1 poor
**Presentation:** 1 poor
**Contribution:** 1 poor
**Rating:** 2
**Confidence:** 3

**Summary:**

This article suggests that the concept of energy-driven network architecture selection in the brain can be applied to artificial neural architecture search (NAS). It proposes redefining the relationship between artificial neurons to include the metabolic cost of computation. The article also discusses the relationship between brain behavioral states and energy management, as well as the theory of dynamic coordination and metabolic optimization in biological systems. It suggests that the optimization of energy-driven systems can be seen as the optimal solution to the complexity of biological systems. The article concludes by proposing an updated NAS strategy that incorporates glial-neuronal ensembles and energy-constrained architecture search.

**Strengths:**

I believe that this kind of interdisciplinary perspective should be encouraged. AI has benefited greatly from biological inspiration and insight, and I commend the authors for their effort. The article proposes that two aspects of biological brains receive higher focus in modern deep learning: the contribution of glial cells and metabolism. I do believe that it is useful to clarify the differences between the biological brain and deep learning on these aspects.

**Weaknesses:**

The article does not sufficiently cover contemporary deep learning literature, which does make links to biological neural networks or their properties. While this is less evident for the inclusion of glial cells, there is substantial literature on network efficiency, including in NAS. It is common practice in NAS to compare architectures not only on error on a test dataset, but also measures of efficiency like parameter count or number of floating point operations. NSGA-Net, for example, explicitly searches using two objectives: one of accuracy and one of efficiency.

Lu, Zhichao, et al. "Nsga-net: neural architecture search using multi-objective genetic algorithm." Proceedings of the genetic and evolutionary computation conference. 2019.

Hardware-aware NAS methods take into account the energy limitations of different hardware, aiming to find a balance between energy efficiency and performance:

Chitty-Venkata, Krishna Teja, and Arun K. Somani. "Neural architecture search survey: A hardware perspective." ACM Computing Surveys 55.4 (2022): 1-36.

A common motivation to compress networks, through pruning or reducing precision, is to reduce their energy use:

Yang, Tien-Ju, Yu-Hsin Chen, and Vivienne Sze. "Designing energy-efficient convolutional neural networks using energy-aware pruning." Proceedings of the IEEE conference on computer vision and pattern recognition. 2017.

The proposed NAS method which takes "metabolic" constraints into account is in the same line, however the article does not acknowledge the existence of this common consideration in the current NAS literature. The claim that "Current AI is based on optimization strategies for information (Imax ) without metabolic constraints (Emin )" is false if we consider measures of floating point operations or computational time on specialized hardware as equivalents to metabolic constraints. The article does not detail how metabolic constraints would be calculated, asking instead "Is Emin for AI just the cost of electricity or also an intrinsic system’s property as exemplified by the theory of dynamic coordination?". Given that measuring energy cost in terms of computational complexity (number of floating point operations) is standard in NAS, the proposed definition of Emin should be clarified in the article.

The article is very short for a NeurIPS submission at less than 5 full pages and could have gone much further in detail both on the proposed ideas and their context in contemporary deep learning.

**Questions:**

What is the appropriate definition for Emin for artificial neural networks? For example, is it FLOPs, computational time on specific hardware, energy consumed during inference passes?

For non-spiking neural networks, how would the influence of glial cells be represented? BatchNorm regulates the activation of a group of neurons, similar to the regulatory behavior of glia. The use of pruning methods during training direct learning towards sparse networks, similar to the energy constraint enforcement of glia. While backpropagation is not considered biologically plausible, the memory of past synaptic modulation stored in the momentum terms of common optimizers like Adam simulate the different timescales of neural action and glial modulation. Are any of these mechanisms sufficient to consider contemporary deep learning architectures as already being "glial-neural nets"?

**Limitations:**

As previously mentioned, the main limitations of this article are the scope of work considered and the lack of detail of the proposed ideas. The proposed ideas should be more fully realized and more firmly grounded in the current literature. The Limitations section on page 5 does not adequately address these limitations.

---

> ### Author Rebuttal · Authors · 2023-08-09
>
> Thank you for the thoughtful comments and criticisms!
>
> First, some general considerations:
>
> When venturing into a new territory there is always a risk of incomplete understanding and the authors admit to this risk with regard to NAS from the onset on Line 14. The work is done in the hope that the NeurIPS audience will find it stimulating and perhaps express interest in pursuing this further, collaboratively with the authors or otherwise. The present comments by all reviewers are proof that this basic assumption is true. It is of course another matter whether this work meets the bar to be presented to the NeurIPS at large. We appreciate the feedback received so far and hope there is a chance that it does, but if not, these comments have already been most helpful.
>
> Second, specific comments on the points made by the reviewers:
>
> We appreciate the assessment that this interdisciplinary perspective is worth pursuing.
>
> Deep learning does not need to follow biological brains, and it has not done so, as this manuscript points out. That is not to suggest that there may be benefits to revisiting some basic assumptions about what a neural net is when we assume it to be biologically inspired. That is the point of the present contribution.
>
> Thank you for the leads on NAS efficiency! In the revised version of the manuscript, these will be included and discussed.
>
> We respectfully disagree with the statement that "Our claim "Current AI is based on optimization strategies for information (Imax ) without metabolic constraints (Emin )" is false if we consider measures of floating point operations or computational time on specialized hardware as equivalents to metabolic constraints". To clarify this further time and/or space complexity seem not as adequate information-theoretical measures of biological metabolic needs, at least not completely. We suggest that the present ideas go beyond being hardware-inspired in that hardware dictates energy limitations and requires efficiency. Rather, energy optimization is proposed to be the fundamental feature of any NAS and the resulting glial-neural net, regardless of the hardware. We propose an approach (Line 128) to derive such an energy-optimized neural net performance metric. An important implication of this approach is that in such an energy landscape V, individual subroutines may well sacrifice some optimization if the total cost is optimized. Whether this is a subtle difference or a qualitative development, hopefully an improvement, on the traditional NAS approach would be interesting to study further.
>
> We do thank the reviewer for the criticism of the Emin definition. Clearly, more work here is needed.
>
> We enjoyed and appreciate the insightful questions around the glial representation in the contemporary net designs. We don't have a brief answer. In fact, in Section 3 we ask the reader to help find it, if one exists. The examples proposed by this reviewer do sound compelling.
> A systematic study of the effects of manipulating these examples on global network behavior would be interesting if designed to ask specifically about the impact of such strategies as representing glial behavior on neural network behavior. Can we reproduce some of the neurological behaviors of autism or schizophrenia, for example? Such a way of thinking could be fruitful as a way of NAS becoming a testing ground for developmental neuroscience, fast-forwarding the developmental clock in silico.
>
> One consideration would be to return to the above notion of the metabolic constraints and suggest that regardless of today's implementation, these system-wide NAS constraints on energy are something not yet implemented in the deep way proposed herein when compared to the observations in developmental neuroscience.

---

### Decision · Program_Chairs · 2023-09-21

**Decision:**

Reject

**Comment:**

The paper proposes a framework for Neural Architecture Search inspired role of glial cells in biological brains. The biggest criticism of the paper is that it more of a conceptual level exposition and does not have either any theoretical insights or any empirical results. Furthermore, reviewers highlight lack of discussion on prior works, as well any discussion that puts the ideas in the context of prior art. All the reviewers recommend rejecting the paper, and I agree with their assessment.